# Abdominal CT organ segmentation by accelerated nnUNet with a coarse to fine strategy

Shoujin Huang[1,2][0000−0001−6094−129X], Lifeng Mei[1], Jingyu Li[3], Ziran Chen[1], Yue Zhang[2], Tan Zhang[4], Xin Nie[2], Kairen Deng[5], and Mengye Lyu[1][0000−0001−5548−8136]

[1] College of Health Science and Environmental Engineering, Shenzhen Technology University, Shenzhen, China
[2] Tencent Music Entertainment, Shenzhen, China
[3] College of Big Data and Internet, Shenzhen Technology University, Shenzhen, China
[4] Sino-German College of Intelligent Manufacturing, Shenzhen Technology University, Shenzhen, China
[5] First Clinical Medical College, Guangdong Medical University, Zhanjiang, China, Corresponding author: `lvmengye@sztu.edu.cn`

**Abstract.** Abdominal CT organ segmentation is known to be challenging. The segmentation of multiple abdominal organs enables quantitative analysis of different organs, providing invaluable input for computer-aided diagnosis (CAD) systems. Based on nnUNet, we develop an abdominal organ segmentation method applicable to both abdominal CT and whole-body CT data. The proposed new training pipeline combines the Kullback–Leibler semi-supervised learning and fully supervised learning, and employs a coarse to fine strategy and GPU accelerated interpolation. Our method achieves a mean Dice Similarity Coefficient (DSC) of 0.873/0.870 and a Normalized Surface Dice (NSD) of 0.911/0.915 on the FLARE 2022 validation/test dataset, with an average process time of 12.27s per case. Overall, we ranked the fifth place in the FLARE 2022 Challenge. The code is available at https://github.com/Solor-pikachu/Infer-MedSeg-With-Low-Resource.

**Keywords:** FLARE 2022 · CT segmentation · Deep Learning.

## 1 Introduction

As a basic subject of medical image analysis, automatic and accurate abdominal organ segmentation from medical images is an essential step for computer-assisted diagnosis, surgery navigation, visual augmentation, radiation therapy, and biomarker measurement systems[9]. In various recent competitions, nnUNet[5] have shown great performance consistently, but its memory consumption and GPU usage lead to huge demand of computing resources, which brings great difficulties on the industrial deployment of this method.

In this paper, we propose an improved training and inference scheme based on nnUNet, and a coarse to fine strategy is added to reduce the computing resources.

The main contributions of this work are summarized as follows:

- A semi-supervised learning algorithm is used to train the model, and 2000 unlabeled CT samples are used to calculate pseudo-labels through the four decoders of the model. Pseudo-labels are used to calculate Kullback–Leibler (KL) divergence loss, and real labels are used to calculate cross entropy and dice loss.
- A coarse to fine strategy based on nnUNet is developed. Compared with the original nnUNet implementation, it achieved significant acceleration with almost no loss of accuracy.
- Unlike the common practice that resizes the CT data to a fixed size, a coarse model with a slide window approach [5] is used to roughly locate abdominal organs in whole-body CT, half-body CT, and abdominal CT, and a fine model is used to perform fine segmentation subsequently.
- The interpolation algorithm of nnUNet is optimized and highly accelerated. For interpolation of large samples of whole-body CT, the time is reduced from 90s to 1s, and the memory consumption is small.

## 2   Method

We propose a method as shown in Figure. 1. We use the coarse model by step=1, to obtain approximate segmentation results from the input CT scan, and then obtain the region of interest(ROI) coordinates of the abdomen from the coarse segmentation. Then we crop the area, and use the fine model for Step=0.5 inference, and finally restore the inference results to the original cropped area according to the ROI coordinates.

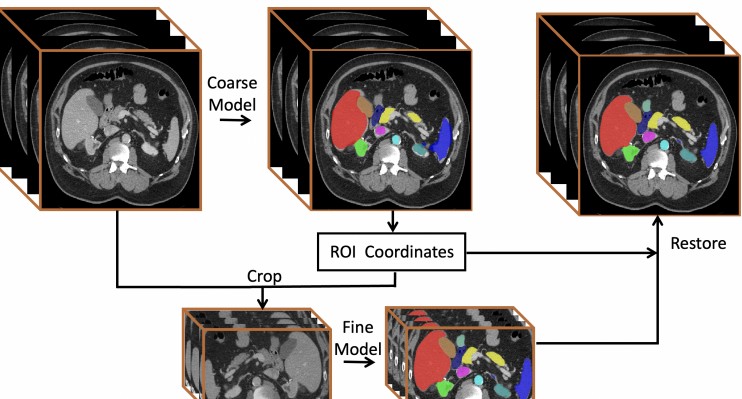

**Fig. 1.** Coarse-to-fine segmentation framework. Coarse and Fine are model inference processes. Crop means cutting the approximate position of the organ from the original image according to the result of coarse segmentation, and Restore means place the result back to the position before cropping.

## 2.1   Preprocessing

We regroup the 50 labeled samples and 2000 unlabeled samples to form two datasets. The first dataset containing all 2050 samples is used to train the model for coarse segmentation, and the second dataset containing only the 50 labeled samples is used to train the model for fine segmentation.

1. In the first dataset, we clip the foreground of the 2050 samples using threshold, and calculate their respective space, max intensity, and min intensity individually. All spaces are resampled to [3.0, 2, 2], and the window width is adjusted to [-325,325]. Last, the intensities of each CT sample are normalized to have a mean of 0 and a variance of 1 using the individual mean and standard deviation.
2. In the second dataset, we adjust the window width to [-325, 325], and resample the spaces to [2, 1.5, 1.5]. The original CT data and label are cropped with a reserved 40mm voxel position, and then the mean and standard deviation are calculated for the population of all samples, and the global mean and standard deviation are used to normalize the intensity of all CT data samples.

## 2.2   Network Architecture

We use a UNet[7] as our model as shown in Figure. 2. The model hyperparameters and the input patch size of [96,128,160] are chosen to satisfy the GPU memory requirement by the FLARE 2022 competition.

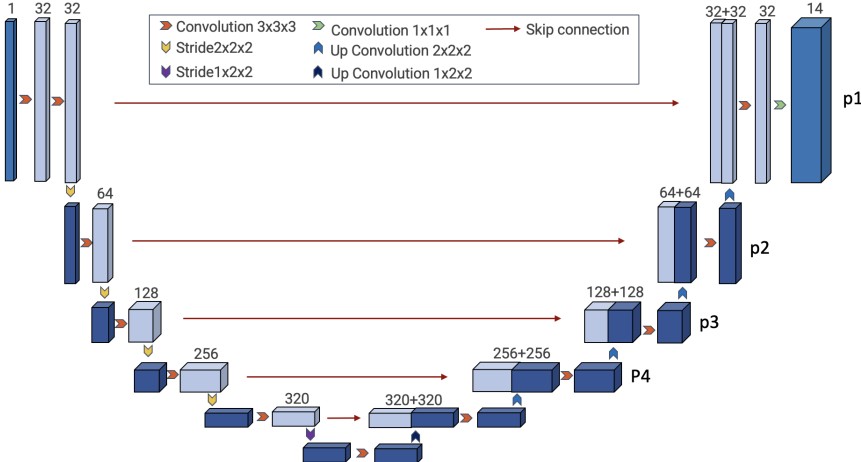

**Fig. 2.** Network architecture. A UNet is used and the outputs of four decoders (P1 to P4) are used to compute loss (see Section 2.3 for details).

### 2.3   Training

**Coarse model training** During the training of the coarse model, we use a semi-supervised algorithm. This idea comes from Xiangde Luo et al.[6], who propose a new uncertainty correction module that enables the framework to gradually learn from meaningful and reliable consensus regions at different scales. For the 50 labeled data, we use cross entropy loss and dice loss to perform supervised learning on four outputs of the decoders (i.e., P1 to P4 shown in Figure. 2).

For the 2000 unlabeled data, we apply the following steps to calculate the loss as shown in Figure. 3:

1. we feed the patch into the model for inference and get four outputs (P1, P2, P3, and P4) from the UNet;
2. we add these four outputs and average them to get the pseudo-label P;
3. the four outputs P1, P2, P3, and P4 of the model are compared with the pseudo-label P to calculate the loss function.

We anticipate that the decoding heads of these four outputs can provide a relatively good pseudo-label by voting on the pseudo-mask. KL-divergence loss is used between the average prediction and the prediction at four scales as the uncertainty measurement.

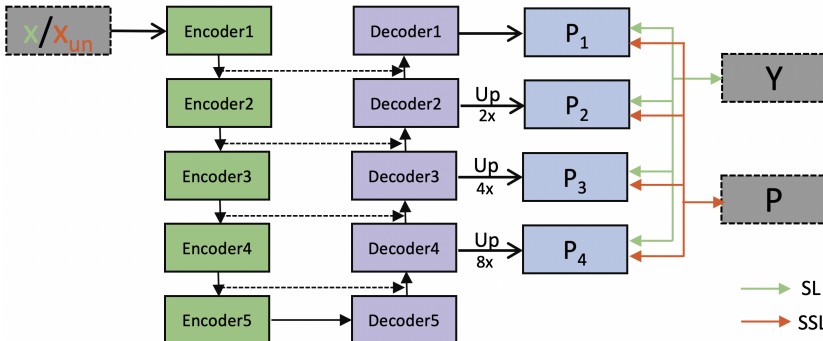

**Fig. 3.** Illustration of the proposed training strategy. Both semi-supervised learning (SSL) and supervised learning (SL) are used. Xun is the unlabeled data, P is the pseudo-label, X is the labeled data, and Y is the label of X.

We give the loss weight of sigmoid to the loss of unlabeled data, so that it can learn less pseudo-labels in the early stage and more in the later stage. We randomly select 40 samples as the training set and 10 samples as the validation set. Here we use Stochastic Gradient Descent (SGD) optimizer with momentum, and set the initial learning rate at 0.01 with poly learning rate decay. A total of 500 epochs of training is done.

**Fine model training** To train the fine model, we initialize the model parameters with the pretrained coarse model. Compared with the randomly initialization, the coarse model has seen more data and improves the convergence speed. As above-mentioned, the cropped 50 gold-standard data are used to fine-tune the model. The optimizer and learning rate decay strategy are the same as in the coarse model training, but the initial learning rate is adjusted to 0.001. A total of 150 epochs of training are done, and in the last 10 epochs, all data augmentations are turned off.

### 2.4   Post-processing

Without post-processing, we found that the model may often mistake bladder and uterus as liver, kidney and stomach, and that the dice scores of the aorta and inferior vena cava were not high, yet the neural network network can often predict the approximate correct location. Thus we perform the post-processing as follows: We find the largest connected area of the aorta and inferior vena cava, then calculate their centroids, and then find the center of these two centroids. We iterate all the connected regions, so to find the distance between the centroid and the center of each connected region. If the centroid is far away beyond a threshold, we delete the connected region. We preserve the largest connected area for all organs separately.

### 2.5   Acceleration on Resize and Argmax Operation

After the neural network finish inference, we need to interpolate the prediction results and restore the original size as input. We find that most of the CT scans are very large in the matrix size, and using the traditional CPU implementation, such as Skimage resize function, is slow, consuming CPU and memory resources. Thus, we change it to Torch's GPU interpolation function, using the trilinear interpolation method. Note that sending the CT array to the GPU at one time may exceed the maximum GPU memory. Therefore, we propose a slice interpolation procedure as shown in Figure. 4. When the output dimension of the sample is [14, z, x, y], it needs to be interpolated to [14, z*2, x*2, y*2], we divide the z-axis into N points, then each slice is [14, z/N, x, y], then we perform GPU-based interpolation on each slice, and finally merge the results of each slice on the Z axis. Similarly, when Argmax operates on the [14, z, x, y] array, we divide the z-axis into N parts again, and then perform the Argmax operation. According to our test, this slicing operation hardly affects the accuracy. Dramatically, the time here is reduced from 90s to 1s after switching to such GPU-based implementation.

## 3   Experiments

### 3.1   Dataset and evaluation measures

The FLARE 2022 dataset is curated from more than 20 medical groups under the license permission, including MSD [8], KiTS [3,4], AbdomenCT-1K [1], and

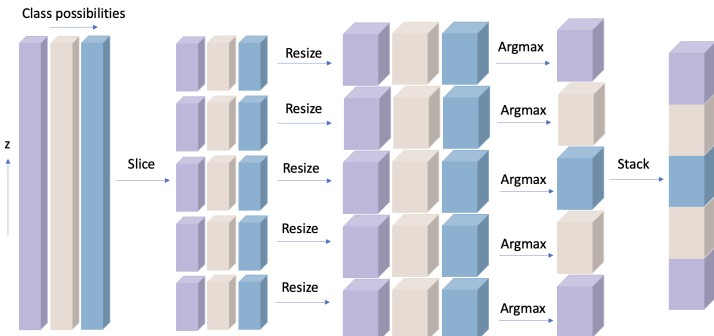

**Fig. 4.** Optimized resize and argmax operation based on slicing.The left represents the low-resolution output of the neural network's likelihood for each organ, and the right represents the segmentation result at the original resolution of the image.

TCIA [2]. The training set includes 50 labelled CT scans with pancreas disease and 2000 unlabelled CT scans with liver, kidney, spleen, or pancreas diseases. The validation set includes 50 CT scans with liver, kidney, spleen, or pancreas diseases. The testing set includes 200 CT scans where 100 cases has liver, kidney, spleen, or pancreas diseases and the other 100 cases has uterine corpus endometrial, urothelial bladder, stomach, sarcomas, or ovarian diseases. All the CT scans only have image information and the center information is not available.

### 3.2  Data Augmentation

During our training process, we introduce the following data augmentation. Gamma change, random Scale change between [0.6, 1.8], random enhancement of contrast, Gaussian blur, random rotation.

### 3.3  Implementation details

**Environment settings** The development environments and requirements are presented in Table 1.

**Training protocols** The Training protocols and details(e.g., batchsize, epoch, optimizer) are presented in Table. 2 and Table. 3.

**Table 1.** Development environments and requirements.

| | |
|---|---|
| Windows/Ubuntu version | Ubuntu 20.04.5 LTS |
| CPU | AMD EPYC 7H12 64-Core Processor |
| RAM | 16×4GB; 2.67MT/s |
| GPU (number and type) | One NVIDIA A100 40G |
| CUDA version | 11.1 |
| Programming language | Python 3.8 |
| Deep learning framework | Pytorch (Torch 1.11.1) |
| Specific dependencies | |
| Link to code | github code |

**Table 2.** Training protocols for the coarse model.

| | |
|---|---|
| Network initialization | "he" normal initialization |
| Batch size | 2 |
| Patch size | 96×128×160 |
| Total epochs | 500 |
| Optimizer | SGD with nesterov momentum ($\mu = 0.99$) |
| Initial learning rate (lr) | 0.01 |
| Lr decay schedule | halved by 200 epochs |
| Training time | 8.5 hours |
| Number of model parameters | 30.79M |
| Number of flops | 225.68G |

**Table 3.** Training protocols for the refine model.

| | |
|---|---|
| Network initialization | pre-train model |
| Batch size | 2 |
| Patch size | 96×128×160 |
| Total epochs | 150 |
| Optimizer | SGD with nesterov momentum ($\mu = 0.99$) |
| Initial learning rate (lr) | 0.001 |
| Lr decay schedule | halved by 150 epochs |
| Training time | 2.5 hours |
| Number of model parameters | 30.79M |
| Number of flops | 225.68G |

## 4    Results and discussion

### 4.1    Quantitative results on validation set

Overall, as shown in Table. 4, our method achieves a mean Dice Similarity Co-efficient (DSC) of 0.8725 and a Normalized Surface Dice (NSD) of 0.9109 on the FLARE 2022 validation dataset, with an average inference time of 15 seconds per case.

**Table 4.** DSC and NSC score in Validation dataset. Liv: liver, RKid: right kidney, Spl: spleen, Pan: pancreas, Aor: aorta, IVC: inferior vena cava, RAG: right adrenal gland, LAG: left adrenal gland, Gall: gallbladder, Eso: esophagus, Sto: stomach, Duo: Duodenum, LKid: left kidney.

| Metric | Liv | RK | Spl | Pan | Aorta | IVC | RAG | LAG | Gall | Eso | Sto | Duo | LKid | Avg. |
|--------|-----|-----|-----|-----|-------|-----|-----|-----|------|-----|-----|-----|------|------|
| DSC | 0.977 | 0.916 | 0.958 | 0.870 | 0.953 | 0.870 | 0.793 | 0.770 | 0.800 | 0.866 | 0.912 | 0.762 | 0.897 | 0.873 |
| NSD | 0.972 | 0.911 | 0.955 | 0.949 | 0.969 | 0.863 | 0.898 | 0.867 | 0.806 | 0.934 | 0.931 | 0.881 | 0.902 | 0.911 |

### 4.2    Quantitative results on final test set

As shown in Table. 5, our method achieves a mean DSC of 0.870 and a NSD of 0.915 on the FLARE 2022 test dataset, with an average inference time of 12.27 seconds per case.

**Table 5.** DSC and NSC score in test dataset. Liv: liver, RKid: right kidney, Spl: spleen, Pan: pancreas, Aor: aorta, IVC: inferior vena cava, RAG: right adrenal gland, LAG: left adrenal gland, Gall: gallbladder, Eso: esophagus, Sto: stomach, Duo: Duodenum, LKid: left kidney.

| Metric | Liv | RK | Spl | Pan | Aorta | IVC | RAG | LAG | Gall | Eso | Sto | Duo | LKid | Avg. |
|--------|-----|-----|-----|-----|-------|-----|-----|-----|------|-----|-----|-----|------|------|
| DSC | 0.981 | 0.946 | 0.948 | 0.821 | 0.958 | 0.867 | 0.8340 | 0.823 | 0.7860 | 0.796 | 0.888 | 0.727 | 0.924 | 0.870 |
| NSD | 0.981 | 0.948 | 0.954 | 0.913 | 0.977 | 0.868 | 0.950 | 0.924 | 0.794 | 0.882 | 0.904 | 0.868 | 0.931 | 0.915 |

### 4.3    Qualitative results on validation

We analyze the samples with relatively good predictions and those with poor predictions. Figure. 6 and Figure. 7 show the results. Samples No. 21 and 23 are good cases, it can be observed that the well-segmented cases have clear organ boundaries. Samples No. 42 and 48 are bad cases, they are often with heterogeneous lesions.

1. It can be seen from the 3D images in Figure. 5 that our neural network can extract masks for most normal organs, but it is difficult to identify the organs with the lesions. In the good cases, the neural network can predict most healthy organs very well, but in the bad cases, the organs with lesions such as kidney tumors are poorly predicted.
2. In Table. 4, Table. 6 and Tabel. 7, our DSC scores are generally lower than the NSD scores on the validation set. Our algorithm misjudges very few regions when predicting, and has high confidence in the segmentation of each organ.

3. From the experimental results in Table. 6 and Tabel. 7, it can be seen that the duodenum, left and right adrenal glands, and inferior venacava have poor DSC scores in the bad cases, but the NSD scores are generally higher than DSC scores. We note that due to deformation and lesions of these organs, the CT HU values of these organs have changed greatly, to which our algorithm is not sensitive. The solution to this deserves further study in the future.

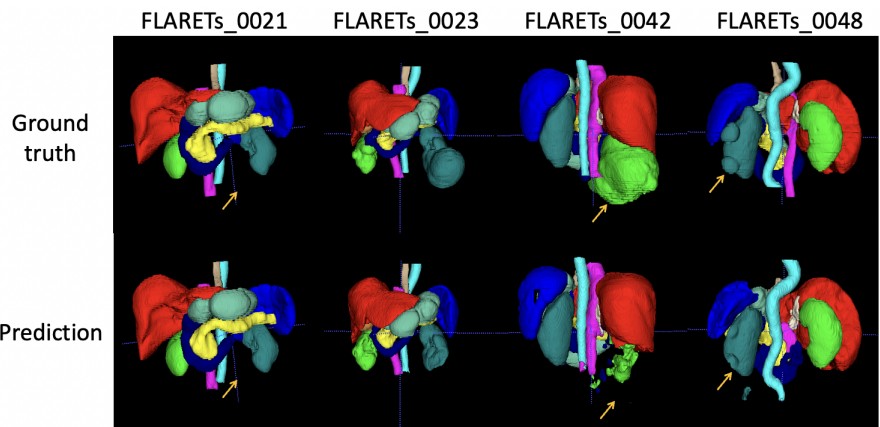

**Fig. 5.** Some representative segmentation results visualized by 3D Viewer

**Table 6.** NSD score of the samples shown in Figure. 6 and 7. Liv: liver, RKid: right kidney, Spl: spleen, Pan: pancreas, Aor: aorta, IVC: inferior vena cava, RAG: right adrenal gland, LAG: left adrenal gland, Gall: gallbladder, Eso: esophagus, Sto: stomach, Duo: Duodenum, LKid: left kidney.

| Case | Liv | RK | Spl | Pan | Aorta | IVC | RAG | LAG | Gall | Eso | Sto | Duo | LKid | Avg. |
|---|---|---|---|---|---|---|---|---|---|---|---|---|---|---|
| FLARES21 | 0.994 | 0.998 | 1 | 0.999 | 1 | 0.959 | 0.988 | 0.995 | 1 | 0.999 | 0.999 | 0.997 | 0.998 | 0.994 |
| FLARES23 | 0.994 | 0.906 | 1 | 0.998 | 1 | 0.895 | 0.937 | 0.965 | 0.896 | 0.991 | 0.974 | 0.915 | 0.611 | 0.930 |
| FLARES42 | 0.968 | 0.062 | 0.869 | 0.881 | 0.951 | 0.926 | 0.738 | 0.743 | 0.974 | 0.999 | 0.916 | 0.606 | 0.956 | 0.815 |
| FLARES48 | 0.9774 | 0.987 | 0.736 | 0.905 | 0.999 | 0.005 | 0.645 | 0.989 | 0 | 0.531 | 0.570 | 0.806 | 0.833 | 0.691 |

**Table 7.** DSC score of the samples shown in Figure. 6 and 7. Liv: liver, RKid: right kidney, Spl: spleen, Pan: pancreas, Aor: aorta, IVC: inferior vena cava, RAG: right adrenal gland, LAG: left adrenal gland, Gall: gallbladder, Eso: esophagus, Sto: stomach, Duo: Duodenum, LKid: left kidney.

| Case | Liv | RK | Spl | Pan | Aorta | IVC | RAG | LAG | Gall | Eso | Sto | Duo | LKid | Avg. |
|---|---|---|---|---|---|---|---|---|---|---|---|---|---|---|
| FLARES21 | 0.990 | 0.985 | 0.991 | 0.931 | 0.982 | 0.951 | 0.911 | 0.900 | 1 | 0.947 | 0.970 | 0.931 | 0.985 | 0.959 |
| FLARES23 | 0.986 | 0.919 | 0.992 | 0.925 | 0.983 | 0.919 | 0.849 | 0.871 | 0.932 | 0.932 | 0.968 | 0.812 | 0.547 | 0.895 |
| FLARES42 | 0.975 | 0.121 | 0.910 | 0.795 | 0.932 | 0.933 | 0.570 | 0.644 | 0.938 | 0.899 | 0.925 | 0.432 | 0.967 | 0.772 |
| FLARES48 | 0.980 | 0.980 | 0.817 | 0.841 | 0.973 | 0 | 0.589 | 0.910 | 0 | 0.480 | 0.524 | 0.546 | 0.897 | 0.657 |

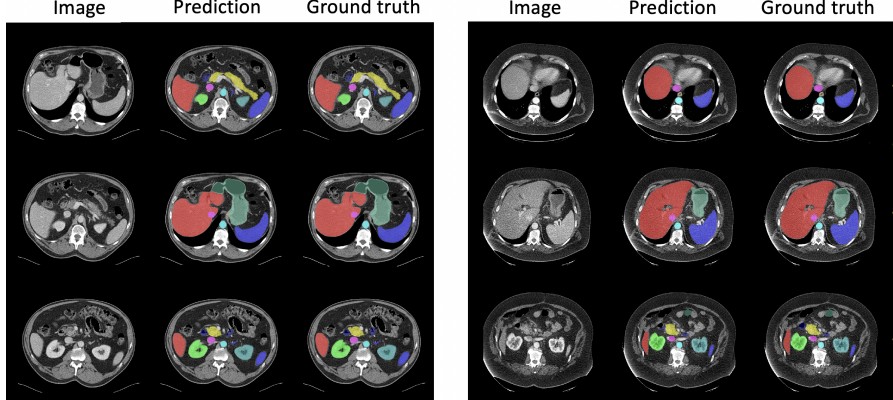

**Fig. 6.** Well segmented cases. Left is sample No.21, and right is sample No.23

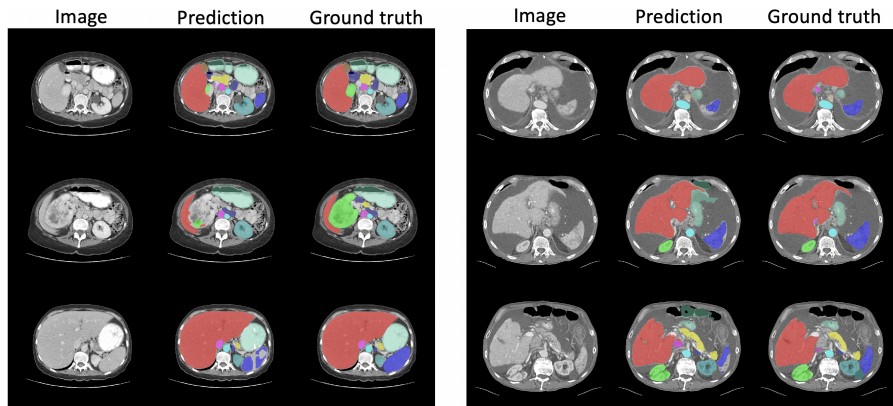

**Fig. 7.** Poorly segmented cases. Left is sample No.42, and right is sample No.48

### 4.4 Tricks for Improvement

As show in Tabel. 8, our segmentation baseline is submitted based on plain nnUNet, and achieves DSC of 0.855 on the validation set. By adding unlabeled data, the DSC reaches 0.866. Further with the proposed coarse-to-fine segmentation, the DSC reaches 0.873 and NSD 0.915 on the validation set.

### 4.5 Two Normalization Strategies

In the first dataset (for coarse segmentation), we normalize the data using the individual normalization method. Because when locating the abdomen on CT scans, there will be full-body CT, half-body CT, and abdominal CT, leading to a big difference between samples. If global normalization is used, information may be erased from the CT intensities of some samples. So we use individual

**Table 8.** Effect of semi-supervised learning and coarse to fine strategy

| Method | Val DSC |
|---|---|
| Baseline | 0.855 |
| Baseline+Unlabeled Data | 0.866 |
| Baseline+Unlabeled Data+Coarse to Fine | 0.873 |

normalization to normalize the data to have a mean of 0 and a variance of 1. In the second dataset (for refining the segmentation), we use global normalization. Because in the first Coarse segmentation, we already obtained the approximate location of the abdomen, we crop the abdomen in the sample.

### 4.6 Effects of Sliding Windows

In the coarse segmentation, we use sliding windows instead of all voxels as input. In fact, we tested performing coarse segmentation without sliding windows in a way similar to the top 1 solution of FLARE 2021[10], and found that the results of half-body CT and whole-body CT were very poor. Through visualization, we noticed that the coarse model didn't segment the approximate position of the abdomen well, due to the fact that whole-body CT and half-body CT are very scarce in the training data. So, it's difficult to improve the segmentation quality even using semi-supervised algorithms in whole-body CT and half-body CT. If the inference was performed in the coarse model without sliding windows, the model usually misidentified a large area as liver or kidney, and these samples were easily connected together, resulting in wrong abdomen locating, and the subsequent fine segmentation may be even worse. The method of using patch sliding window inference can reduce the occurrence of this problem.

## 5   Conclusion

In this paper, we propose an algorithm based on nnUNet to develop an abdominal organ segmentation method that can handle both abdominal CT and whole-body CT, through coarse-to-fine segmentation scheme, using semi-supervised algorithms. Quantitatively evaluated, the method achieves an average DSC of 0.873, and a NSD of 0.911 with an average process time of 15s per case in the validation dataset. Also we achieve an average DSC of 0.870, and a NSD of 0.915 with an average process time of 12.27s per case in the test dataset.

**Acknowledgements**  The authors of this paper declare that the segmentation method they implemented for participation in the FLARE 2022 challenge has not used any pre-trained models nor additional datasets other than those provided by the organizers. The proposed solution is fully automatic without any manual intervention. We thank to the timely help given by Bingding Huang in supporting

GPU machine, Sixin Liu in supporting word spelling and grammar correction. This study is supported in part by Natural Science Foundation of Top Talent of Shenzhen Technology University (Grants No. 20200208 to Lyu, Mengye and No. GDRC202134 to Li, Jingyu), and the National Natural Science Foundation of China (Grant No. 62101348 to Lyu, Mengye).

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
