# OpenReview forum: "Abdominal CT organ segmentation by accelerated nnUNet with a coarse to fine strategy"
_MICCAI.org/2022/Challenge/FLARE_

### Official Review · Reviewer_ASWo · 2022-09-16
**Abdominal CT organ segmentation by accelerated nnUNet with a coarse to fine strategy**

**Rating:** 8
**Confidence:** 4

**Review:**

Strengths: The proposed method achieves efficient and effective semi-supervised learning with a mean DSC of 0.8725 and a mean inference time of 15 s. Especially, Pytorch-and-GPU-based interpolation for resizing and optimizing the argmax operation dramatically reduced the inference time.

Weaknesses: Efficiency results other than the inference time are not presented explicitly, although Section 2.2 states that the proposed method satisfies the GPU memory requirement of the FLARE22 challenge.

---

> ### Author Response · Authors · 2022-10-14
> **Reply**
>
> Thanks for your advice. We have rewrite the Efficiency results on our abstract, section 4.1 and 4.2.

---

### Official Review · Reviewer_LWJd · 2022-09-16
**Great work and achieve a really fast and fine segmentation!**

**Rating:** 9
**Confidence:** 4

**Review:**

Pros:
1) This work segment the abdominal organs in a coarse to fine manner and achieving 0.8725 mean DSC.
2) They resampled the data to [3, 2, 2] mm voxel-size for coarse segmentation and [2, 1.5, 1.5] mm for fine segmentation to reduce the resource consume.
3) They also do some optimization for acceleration which reduce the time greatly!


Cons:
1) They did't show the exactly loss for semi-suprevised learning and unlabeled data.

---

> ### Author Response · Authors · 2022-10-14
> **Reply**
>
> Thanks for your advice. Due to the page limit, we did not draw the separate losses, however, similar information can be seen from the ablation study results in Table 8.

---

### Official Review · Reviewer_v8ST · 2022-09-16
**Unlabeled data might also be used for the fine stage model training**

**Rating:** 9
**Confidence:** 4

**Review:**

Pros: 1. Interpolation method is optimized based on nnUNet so the inference efficiency is highly accelerated. 2. Unlabeled data is used by semi-supervised training in coarse model which has a ~1% boot in DSC.

Cons: Unlabeled data might also be used for the fine stage model training.

---

> ### Author Response · Authors · 2022-10-13
> **Reply**
>
> Thanks for your advice. We initial the fine model with the pre-trained model from the coarse model when training.

---

### Official Review · Reviewer_Bojc · 2022-09-19
**The article proposes a practical and efficient method based on the proposition of FLARE2022, which achieves a good performance of abdominal segmentation.**

**Rating:** 8
**Confidence:** 5

**Review:**

Pros: 1. This job used a multi-scale strategy to obtain the pseudo-label and the uncertainty correction enabled a more reliable consistency. This semi-supervised algorithm improved the performance by using the unlabeled data and has been proved by the experimental result.
2. A coarse to fine strategy based on nnUNet has achieved acceleration which shows its consideration for the fast inference.
3. The article has a good presentation of methods and details
Cons:  1. The number of model parameters and flops might not meet the requirements of low resource.
2. The high accuracy mainly come from the pre-processing and coarse-to-fine framework, while the contribution of semi-supervised algorithm was small, denoting that it has not fully use the unlabeled data.

---

> ### Author Response · Authors · 2022-10-14
> **Reply**
>
> Thanks for your advice. We will do better in the future.

---

### Official Review · Reviewer_WPTp · 2022-09-19
**Review of Abdominal CT organ segmentation by accelerated nnUNet with a coarse to fine strategy**

**Rating:** 8
**Confidence:** 4

**Review:**

The authors give detailed description of their method and conduct fully experiments to achieve promising performance.

---

> ### Author Response · Authors · 2022-10-14
> **Reply**
>
> Thanks for your advice.

---

### Official Review · Reviewer_HXtB · 2022-09-20
**Great network design and SSL method**

**Rating:** 9
**Confidence:** 4

**Review:**

Advantages: 1. Great two-phase network with outstanding performance and efficiency. 2. Rational SSL method with performance improvement.
Disadvantages: Fig.5 and typography should be checked.

---

> ### Author Response · Authors · 2022-10-14
> **Reply**
>
> Thanks for your advice. We will do better in the future.

---

### Official Review · Reviewer_QcDT · 2022-09-21
**Good work!**

**Rating:** 9
**Confidence:** 4

**Review:**

The method achieves a mean Dice Similarity Coefficient (DSC) of 0.8725 and a Normalized Surface Dice (NSD) of 0.9109 on the FLARE2022 validation dataset, with an average inference time of 15 seconds per case.This is an excellent result.

---

> ### Author Response · Authors · 2022-10-14
> **Reply**
>
> Thanks for your advice. We will do better in the future.

---

### Meta-Review · Program_Chairs · 2022-09-28

**Recommendation:** Minor Revision
**Confidence:** 5

**Metareview:**

Nice paper. Please address the reviewers' comments in the revised manuscript.